# Evaluation of the Specific Energy Consumption of Sea Water Reverse Osmosis Integrated with Membrane Distillation and Pressure–Retarded Osmosis Processes with Theoretical Models

**DOI:** 10.3390/membranes12040432

**Published:** 2022-04-16

**Authors:** Shao-Chi Tsai, Wei-Zhi Huang, Geng-Sheng Lin, Zhen Wang, Kuo-Lun Tung, Ching-Jung Chuang

**Affiliations:** 1R&D Center for Membrane Technology, Department of Chemical Engineering, Chung Yuan Christian University, Taoyuan 320, Taiwan; angela10221248@gmail.com; 2Department of Chemical Engineering, National Taiwan University, Taipei 106, Taiwan; r07524009@g.ntu.edu.tw (W.-Z.H.); r05524113@ntu.edu.tw (G.-S.L.); d07524021@ntu.edu.tw (Z.W.); 3Water Innovation, Low Carbon and Environmental Sustainability Research Center (WInnER), National Taiwan University, Taipei 106, Taiwan; 4Advanced Research Center for Green Materials Science and Technology, National Taiwan University, Taipei 106, Taiwan

**Keywords:** reverse osmosis, pressure-retarded osmosis, membrane distillation, process integration, specific energy consumption

## Abstract

In this study, theoretical models for specific energy consumption (SEC) were established for water recovery in different integrated processes, such as RO-PRO, RO-MD and RO-MD-PRO. Our models can evaluate SEC under different water recovery conditions and for various proportions of supplied waste heat. Simulation results showed that SEC in RO increases with the water recovery rate when the rate is greater than 30%. For the RO-PRO process, the SEC also increases with the water recovery rate when the rate is higher than 38%, but an opposite trend can be observed at lower water recovery rates. If sufficient waste heat is available as the heat source for MD, the integration of MD with the RO or RO-PRO process can significantly reduce SEC. If the total water recovery rate is 50% and MD accounts for 10% of the recovery when sufficient waste heat is available, the SEC values of RO, RO-PRO, RO-MD and RO-MD-PRO are found to be 2.28, 1.47, 1.75 and 0.67 kWh/m^3^, respectively. These critical analyses provide a road map for the future development of process integration for desalination.

## 1. Introduction

Climate change and water scarcity are the two pervasive problems posing serious threats to people around the world [1]. To solve these problems, seawater desalination technology for potable water has been rapidly developed. Reverse osmosis (RO) is currently the most common membrane process for desalination since high-performance RO membranes and modules have been well established. However, this process requires high electric energy consumption and the discharge of concentrated brine solution, which have confined the application of RO systems for years [2,3,4,5,6,7]. Thus, developing methods to reduce RO energy consumption and brine disposal is crucial for process efficiency and sustainability. One of the techniques that has received extensive attention is process integration, or the combination of RO with pressure-retarded osmosis (PRO) and/or membrane distillation (MD). This integrated approach can reduce the energy requirement for water recovery and enhance the water recovery rate [8].

PRO is an energy-harvesting process that converts the osmotic pressure of a saline solution to hydraulic pressure. In a PRO system, water from a low-salinity solution (i.e., the feed solution, FS) is transported through the membrane to a high-salinity solution (i.e., the draw solution). This chemical potential difference can later be converted to electrical energy through hydroturbines [9,10,11,12] or mechanical energy through a pressure exchanger (PX). Additionally, MD is a promising technology for treating saline water and wastewater with high rejection factors. In an MD system, vapor molecules are transferred through a microporous hydrophobic membrane, and this process is driven by the partial vapor pressure difference induced by the temperature gradient across the membrane. This system relies on a thermally driven separation process; therefore, its performance is less sensitive to the salt concentration in the feed than the RO process [13,14,15,16,17,18].

Currently, there are five prevailing process integration schemes for desalination. The first is the simple RO process. Although it can produce potable water, most of the pressure energy leaves the membrane module with the brine. The second is the RO process combined with energy recovery devices (ERDs), such as PXs [19,20,21]. This process is designed to recover the residual energy of the brine. The third is the RO-PRO process [5,22,23]. This process can significantly reduce specific energy consumption (SEC) in the RO process. In this process, seawater flows into the RO system and is separated into pure water and brine. The RO brine is the input into the PRO system as the DS, which is later diluted, where energy is generated. The diluted brine stream from PRO can be discharged into the ocean without the concern of deteriorating marine habitats. The fourth approach is the RO-MD process. This process can increase water recovery [24]. The final method is the RO-MD-PRO process. This process can not only reduce SEC but can also increase water recovery [25].

To date, several research groups have evaluated the feasibility of integrating RO, MD, and PRO processes with software simulations. Prante et al. investigated RO and RO-PRO systems at a 50% RO water recovery rate. Their simulation results indicated that the SECs of RO and RO-PRO are 2.0 and 1.2 kWh/m^3^, respectively, i.e., a 40% SEC reduction is achieved after RO is integrated with PRO [26]. Wan et al. studied models of RO without PX, RO with PX, and RO-PX-PRO systems. The results showed that when the RO water recovery rate is 25%, the SECs of these three systems are 5.51, 1.79, and 1.08 kWh/m^3^, respectively; moreover, when the RO water recovery rate is increased to 50%, the SECs of these three systems is 4.13, 2.27, and 1.14 kWh/m^3^, respectively [27]. Kim et al. discussed the SEC of RO and the RO-MD-PRO system, and they found that the SEC is 1.914 kWh/m^3^ for RO with a water recovery rate of 49.7% and 1.61 kW/m^3^ for RO-MD systems if (i) a heat source is supplied by waste heat and (ii) the MD water recovery rate is 5% [25]. Moreover, Ruiz-Garcia et al. analyzed the effect of different feed spacer geometries on the SWRO spiral-wounded membrane module. Their simulation results indicated that the longer the pressure vessel, the higher the influence of the feed spacer geometry on SEC [28].

Previous studies indicated that after integrating RO with MD and/or PRO, SEC can be reduced; however, these studies mainly focused on situations with specific RO and MD water recovery rates. It is acknowledged that (i) the water recovery rate is the main parameter that determines the minimum operating pressure of an RO system; (ii) the MD water recovery rate and percentage of waste heat from the heat source in the MD system affect the required operating energy; and (iii) the total water recovery of the integrated process affects the feed concentration of the PRO draw solution, which in turn affects the maximum power density generated by the PRO system. Therefore, the water recovery rate of RO and MD and the proportion of waste heat for the MD heat source have a crucial influence on SEC in the integrated process. Herein, we established theoretical models based on energy consumption in fluid transport, fluid mechanical energy recovery, the MD thermal energy demand and PRO energy production to analyze the SEC of RO, RO-PRO, RO-MD and RO-MD-PRO systems at different water recovery rates and different percentages of waste heat supplied for the MD system.

## 2. Materials and Methods

In this study, Equation (1) is used to indicate the energy consumed per unit of water production [2,3].
(1)$$\mathrm{SEC}=\frac{\mathrm{power}\;\mathrm{consumption}\;(\mathrm{kW})}{\mathrm{water}\;\mathrm{production}\;\mathrm{rate}\;(\frac{m^{3}}{h})}$$

This study integrated RO with other unit operations, such as ERD, MD, and PRO, with the aim of analyzing SEC variations in different integrated processes.

### 2.1. SEC of the RO Process

A schematic diagram of the stand-alone RO process is shown in Figure 1. Since the suspended particles, microbes, and organic and inorganic matter in seawater can block the RO membrane and lead to fouling, the raw seawater feed solution is often pretreated with UF/MF prior to RO. The SEC for pretreatment per unit of RO permeate (SEC_Pre_) is described in Equation (2).
(2)$${\mathrm{SEC}}_{\mathrm{Pre}}=\frac{{\mathrm{SEC}}_{\mathrm{pre},\mathrm{sw}}Q_{\mathrm{SW}}}{Q_{P,\mathrm{RO}}}=\frac{{\mathrm{SEC}}_{\mathrm{pre},\mathrm{sw}}}{Y_{r}}$$
where SEC_Pre,sw_ is the pretreatment energy consumption per unit seawater feed solution, Q_SW_ is the volumetric flow rate of the seawater feed solution (m^3^/h), Q_P,RO_ is the RO permeate flow rate (m^3^/h), and $Y_{r}\left(=\frac{Q_{P,\mathrm{RO}}}{Q_{\mathrm{SW}}}\right)$ is the RO water recovery rate.

Glueckstern and Priel suggested that when ultrafiltration is used prior to RO, SEC_Pre,sw_ is 0.095 kWh/m^3^ [29]. Pretreated seawater is pressurized via a high-pressure pump (HP) to the RO operating pressure (P_RO_) and then enters the RO unit. The minimum operating pressure required for RO is the osmotic pressure of the brine. In this study, the RO operating condition is set equal to the minimum operating pressure at each water recovery rate (i.e., osmotic pressure of brine in RO) for determining the SEC. The SEC of the stand-alone RO process (SEC_RO_) was evaluated by Equation (3) [2,3].
(3)$${\mathrm{SEC}}_{\mathrm{RO}}=\frac{\pi_{\mathrm{sw}}R_{t}}{\eta_{p}Y_{r}\left({1-Y}_{r}\right)}$$
where π_sw_ is the osmotic pressure of the feed solution (bar), R_t_ is the salt rejection rate of the membrane, and η_p_ is the HP efficiency. The osmotic pressure can be calculated by the van ’t Hoff equation as below [30]:(4)π=iCRT
where π is the osmotic pressure (bar), i is the van ’t Hoff factor (i = 1.9 when the solute is NaCl [30]), C is the salt concentration (M), R is the ideal gas constant, and T is the temperature (K). In this study, the temperature is set to 298 K, unless a different value is specified.

The RO-ERD process is presented in Figure 2. ERD is utilized to recover the remaining mechanical energy in the brine solution; herein, a PX was selected as the ERD. After seawater pretreatment, part of the feed solution enters the RO unit via HP, and the other part enters the RO unit via the PX and booster pump (BP). The brine solution is then discharged after the available brine energy is transferred via the PX at atmospheric pressure. The SEC of the RO-ERD process (SEC_RO-ERD_) can be calculated as in Equation (5) [2,3]:(5)$${\mathrm{SEC}}_{\mathrm{RO}-\mathrm{ERD}}=\frac{\pi_{\mathrm{sw}}R_{t}\left({1-\eta }_{\mathrm{PX}}\left({1-Y}_{r}\right)\right)}{\eta_{p}Y_{r}\left({1-Y}_{r}\right)}$$
where η_PX_ is the PX efficiency.

### 2.2. SEC of the RO-PRO Process

Figure 3 shows the flow chart of the RO-PRO process. The brine solution, which served as the DS for the PRO unit, is discharged from the RO system and releases pressure via the PX1 to PRO operating pressure (P_PRO_). Previous studies often used low-saline water as the FS for PRO; however, in practice, urban wastewater should be considered as the PRO FS [27]. In this study, a 0.01-M NaCl solution is used to simulate the FS of the PRO unit. Considering the environmental impact after a large amount of brine discharge, the RO brine is assumed to be diluted to the original seawater concentration after passing through the PRO unit. The SEC of RO-PRO (SEC_RO-PRO_) can be calculated as in Equation (6).
(6)$${\mathrm{SEC}}_{\mathrm{RO}-\mathrm{PRO}}=\frac{{\dot{W}}_{\mathrm{pump},\mathrm{RO}-\mathrm{PRO}}}{\eta_{p}Q_{P,\mathrm{RO}}}=\frac{{\dot{W}}_{\mathrm{pump},\mathrm{RO}}-{\dot{W}}_{\mathrm{pump},\mathrm{PX}1}-{\dot{W}}_{\mathrm{pump},\mathrm{PX}2}}{\eta_{p}Q_{P,\mathrm{RO}}}$$
where ${\dot{W}}_{\mathrm{pump},\mathrm{RO}-\mathrm{PRO}}$ and ${\dot{W}}_{\mathrm{pump},\mathrm{RO}}$ are the energy consumption levels of the HP in RO-PRO and stand-alone RO processes, respectively. ${\dot{W}}_{\mathrm{pump},\mathrm{RO}}$ can be calculated in Equation (7).
(7)$${\dot{W}}_{\mathrm{pump},\mathrm{RO}}=P_{\mathrm{RO}}{\times Q}_{\mathrm{SW}}$$
${\dot{W}}_{\mathrm{pump},\mathrm{PX}1}$ is the mechanical energy recovered by PX1, ${\dot{W}}_{\mathrm{pump},\mathrm{PX}2}$ is the energy recovered by PX2, PRO is the RO operating pressure, and Q_sw_ is the volumetric flow rate at the seawater inlet.

Previous studies indicated that the maximum power density can be obtained when the operating pressure equals half of the transmembrane osmotic pressure [31]. Thus, P_PRO_ is set accordingly in Equation (8).
(8)$$P_{\mathrm{PRO}}=\frac{1}{2}(\pi_{\mathrm{RO}-\mathrm{PRO},\mathrm{DS}}{-\pi }_{\mathrm{PRO},\mathrm{FS}})$$
where $\pi_{\mathrm{RO}-\mathrm{PRO},\mathrm{DS}}$ and $\pi_{\mathrm{PRO},\mathrm{FS}}$ are the osmotic pressures of the DS and FS, respectively. The recovery of the mechanical energy of the RO brine after depressurization from P_RO_ to P_PRO_ via PX1 (${\dot{W}}_{\mathrm{pump},\mathrm{PX}1}$) can be evaluated by Equation (9).
(9)$${\dot{W}}_{\mathrm{pump},\mathrm{PX}1}=\eta_{\mathrm{PX}}{(P}_{\mathrm{RO}}{-P}_{\mathrm{PRO}}{)Q}_{B,\mathrm{RO}}$$
where Q_B,RO_ is the RO brine flow rate. The recovery of mechanical energy at the PRO DS outlet after depressurization from P_PRO_ to atmospheric pressure via PX2 (${\dot{W}}_{\mathrm{pump},\mathrm{PX}2}$) can be evaluated by Equation (10).
(10)$${\dot{W}}_{\mathrm{pump},\mathrm{PX}2}=\eta_{\mathrm{PX}}\left(P_{\mathrm{PRO}}{-0)(Q}_{B,\mathrm{RO}}{+Q}_{P,\mathrm{PRO}}\right)=\eta_{\mathrm{PX}}P_{\mathrm{PRO}}Q_{\mathrm{PRO},O}$$
where Q_PRO,O_ and Q_P,PRO_ are the flow rate at the PRO DS outlet and permeation rate through the PRO membrane, respectively. After combining Equation (6) and (10), the SEC of the RO-PRO process can be presented as shown in Equation (11).
(11)$${\mathrm{SEC}}_{\mathrm{RO}-\mathrm{PRO}}{=\mathrm{SEC}}_{\mathrm{RO}-\mathrm{ERD}}-\frac{\eta_{\mathrm{PX}}P_{\mathrm{PRO}}{(Q}_{P,\mathrm{PRO}}/Q_{\mathrm{SW}})}{\eta_{p}\left(Y_{r}\right)}$$

### 2.3. SEC of the RO-MD Process

Figure 4 illustrates the RO-MD process. RO brine is first depressurized to P_MD_ via PX3 and then transported to the MD unit. The SEC of RO-MD (${\mathrm{SEC}}_{\mathrm{RO}-\mathrm{MD}}$) and the mechanical energy recovered via PX3 (${\dot{W}}_{\mathrm{pump},\mathrm{PX}3}$) can be presented by Equations (12) and (13), respectively.
(12)$${\mathrm{SEC}}_{\mathrm{RO}-\mathrm{MD}}=\frac{{\dot{W}}_{\mathrm{pump},\mathrm{RO}-\mathrm{MD}}}{\eta_{p}\left(Q_{P,\mathrm{RO}}{+Q}_{P,\mathrm{MD}}\right)}=\frac{{\dot{W}}_{\mathrm{pump},\mathrm{RO}}-{\dot{W}}_{\mathrm{pump},\mathrm{PX}3}}{\eta_{p}\left(Q_{P,\mathrm{RO}}{+Q}_{P,\mathrm{MD}}\right)}$$
(13)$${\dot{W}}_{\mathrm{pump},\mathrm{PX}3}=\eta_{\mathrm{PX}}{(P}_{\mathrm{RO}}{-P}_{\mathrm{MD}}{)Q}_{B,\mathrm{RO}}$$
where Q_P,MD_ is the MD permeate flow rate and Q_B,MD_ is the brine flow rate discharged from MD. Assuming there is sufficient waste heat for the MD unit, ${\mathrm{SEC}}_{\mathrm{RO}-\mathrm{MD}}$ described in Equation (12) can be converted to the form shown in Equation (14).
(14)$${\mathrm{SEC}}_{\mathrm{RO}-\mathrm{MD}}=\frac{\pi_{\mathrm{sw}}R_{t}\left({1-\eta }_{\mathrm{PX}}\left({1-Y}_{r}\right)\right){+\eta }_{\mathrm{PX}}P_{\mathrm{MD}}{\left({1-Y}_{r}\right)}^{2}}{\eta_{p}\left({1-Y}_{r}\right){(Y}_{r}{+Y}_{m})}$$
where $Y_{m}\left(=\frac{Q_{P,\mathrm{MD}}}{Q_{\mathrm{SW}}}\right)$ is the MD water recovery rate. If the waste heat supply for MD is not sufficient and a supplementary heat source is then required, the MD thermal energy consumption (${\mathrm{SEC}}_{\mathrm{Thermal}-\mathrm{MD}}$) and total SEC of RO-MD (${\mathrm{SEC}}_{\mathrm{RO}-\mathrm{MD}}^{\mathrm{Thermal}}$) can be expressed as shown in Equations (15) and (16).
(15)$${\mathrm{SEC}}_{\mathrm{Thermal}-\mathrm{MD}}=\left(1-X\right)\frac{\rho_{P,\mathrm{MD}}Q_{P,\mathrm{MD}}\Delta H_{\mathrm{vap}}}{\mathrm{EE}\left(Q_{P,\mathrm{RO}}{+Q}_{P,\mathrm{MD}}\right)}$$
(16)$${\mathrm{SEC}}_{\mathrm{RO}-\mathrm{MD}}^{\mathrm{Thermal}}{=\mathrm{SEC}}_{\mathrm{RO}-\mathrm{MD}}{+\mathrm{SEC}}_{\mathrm{Thermal}-\mathrm{MD}}$$
where X is the proportion of waste heat supplied to the total thermal energy consumption of MD and EE is the energy efficiency of MD, which is defined as the percentage of the thermal energy associated with liquid evaporation [13]. $\rho_{P,\mathrm{MD}}$ is the density of water, and $\Delta H_{\mathrm{vap}}$ is the enthalpy of vaporization.

### 2.4. SEC of the RO-MD-PRO Process

The power density of PRO can be enhanced by increasing the DS concentration; therefore, utilizing the high-concentration RO-MD brine discharge as the PRO DS can improve power production. Figure 5 is the schematic diagram of the integrated RO-MD-PRO process. MD brine is pressurized in the PRO unit as the DS. It is assumed that PRO DS will be diluted to the initial seawater concentration and that the mechanical energy will be transferred to the RO feed solution via PX5. The SEC of RO-MD-PRO (${\mathrm{SEC}}_{\mathrm{RO}-\mathrm{MD}-\mathrm{PRO}}$) and the recovery mechanical energy from PX4 (${\dot{W}}_{\mathrm{pump},\mathrm{PX}4-\mathrm{recover}}$) can be expressed as shown in Equations (17) and (18).
(17)$${\mathrm{SEC}}_{\mathrm{RO}-\mathrm{MD}-\mathrm{PRO}}=\frac{{\dot{W}}_{\mathrm{pump},\mathrm{RO}-\mathrm{MD}-\mathrm{PRO}}}{\eta_{p}\left(Q_{P,\mathrm{RO}}{+Q}_{P,\mathrm{MD}}\right)}=\frac{{\dot{W}}_{\mathrm{pump},\mathrm{RO}}-{\dot{W}}_{\mathrm{pump},\mathrm{PX}4-\mathrm{recover}}+{\dot{W}}_{\mathrm{pump},\mathrm{PX}4-\mathrm{reuse}}-{\dot{W}}_{\mathrm{pump},\mathrm{PX}5}}{\eta_{p}\left(Q_{P,\mathrm{RO}}{+Q}_{P,\mathrm{MD}}\right)}$$
(18)$${\dot{W}}_{\mathrm{pump},\mathrm{PX}4-\mathrm{recover}}=\eta_{\mathrm{PX}}{(P}_{\mathrm{RO}}{-P}_{\mathrm{MD}}{)Q}_{B,\mathrm{RO}}$$
(19)$${\dot{W}}_{\mathrm{pump},\mathrm{PX}4-\mathrm{reuse}}=\eta_{\mathrm{PX}}{(P}_{\mathrm{PRO}}{-P}_{\mathrm{MD}}{)Q}_{B,\mathrm{MD}}$$
(20)$${\dot{W}}_{\mathrm{pump},\mathrm{PX}5}=\eta_{\mathrm{PX}}\left(P_{\mathrm{PRO}}{-0)(Q}_{B,\mathrm{MD}}{+Q}_{P,\mathrm{PRO}}\right)$$
where ${\dot{W}}_{\mathrm{pump},\mathrm{PX}4-\mathrm{reuse}}$ is the mechanical energy transferred from the RO brine to PRO DS via PX4, and ${\dot{W}}_{\mathrm{pump},\mathrm{PX}5}$ is the mechanical energy recovered from the PRO DS outlet as the DS depressurized from PPRO to atmospheric pressure via PX5.

By combining Equations (17)–(20) and assuming that there is sufficient waste heat to meet the thermal energy requirement of MD (X = 1), the SEC of the RO-MD-PRO process can be obtained.
(21)$${\mathrm{SEC}}_{\mathrm{RO}-\mathrm{MD}-\mathrm{PRO}}\left(X=1\right)=\frac{\pi_{\mathrm{sw}}R_{t}\left({1-\eta }_{\mathrm{PX}}\left({1-Y}_{r}\right)\right)}{\eta_{p}\left({1-Y}_{r}\right){(Y}_{r}{+Y}_{m})}+\frac{\eta_{\mathrm{PX}}P_{\mathrm{MD}}Y_{m}{-\eta }_{\mathrm{PX}}P_{\mathrm{PRO}}{(Q}_{P,\mathrm{PRO}}/Q_{\mathrm{SW}})}{\eta_{p}{(Y}_{r}{+Y}_{m})}$$

### 2.5. The Fractional Energy Savings

Based on the RO-ERD process, the fractional energy savings (FES) in water production based on hybrid processes can be defined as shown in Equation (22).
(22)$$\mathrm{FES}=\frac{{\mathrm{SEC}}_{\mathrm{RO}-\mathrm{ERD}}{-\mathrm{SEC}}_{\mathrm{integrated}\;\mathrm{process}}}{{\mathrm{SEC}}_{\mathrm{RO}-\mathrm{ERD}}}\times 100\%$$

When ${\mathrm{SEC}}_{\mathrm{integrated}\;\mathrm{process}}$ is zero, the FES equals 100%. Under these circumstances, energy consumption and energy generation are equal, i.e., no further energy supply is required for this process. However, when the FES is greater than 100%, the energy generation is larger than the energy consumption, and ${\mathrm{SEC}}_{\mathrm{integrated}\;\mathrm{process}}$ for this process is negative, and vice versa.

## 3. Results and Discussion

Herein, the SEC of RO, RO-PRO, RO-MD, and RO-MD-PRO were analyzed under different operating conditions. These calculated SECs were further compared with data from previous research articles.

### 3.1. RO Process

Table 1 lists the parameters used in stand-alone RO and/or RO-ERD process simulations. Figure 6 shows the variations in SEC_RO_ and SEC_RO-ERD_ at different water recovery rates. The SEC of the pretreatment unit, SEC_pre_, is also shown in Figure 6. As the water recovery rate decreases, the amount of seawater to be pretreated increases, which leads to an increase in SEC_pre_. SEC_RO_ is much higher than SEC_pre_; therefore, the influence of SEC_pre_ on SEC is small in the stand-alone RO process. The SEC_RO_ reaches a minimum value of 4.0 kWh/m^3^ when the water recovery rate is 51%. Additionally, SEC_RO-ERD_ is lower than SEC_RO_; hence, the addition of the pretreatment unit has a pronounced influence on the total SEC. In the RO-ERD process, SEC is maintained at approximately 2.0 kWh/m^3^ when the water recovery rate ranges from 27–36%.

### 3.2. RO-PRO Process

The PRO unit provides both power generation and a reduction in the brine discharge concentration. Figure 7 shows the energy consumption of RO and energy generation of PRO in the RO-PRO process. When the water recovery rate is higher than 29%, both power generation and energy consumption increase as the recovery rate increases; however, the rate of increase of the latter is significantly larger. The results show that RO energy consumption is more dominant than PRO energy generation in the RO-PRO process.

Figure 8 shows a comparison of SEC between the RO-ERD and RO-PRO processes. When the water recovery rate is low, the SEC of these two processes decreases as the water recovery rate increases. Notably, the increased water recovery rate will reduce the amount of seawater required, thereby decreasing SEC_pre_. However, when the water recovery rate is high, SEC increases as the water recovery rate increases. Since the RO operating pressure increases as the water recovery rate increases, HP energy consumption results in an increase in the total SEC. When the RO water recovery rate is 30%, the RO-ERD process yields a minimum SEC of 1.97 kWh/m^3^, and the RO-PRO process has a minimum SEC of 1.33 kWh/m^3^ when the RO water recovery rate is 38% [32]. Comparing the two processes, RO-PRO has a lower SEC due to the energy generation by PRO and obtains a higher water recovery rate at the minimum SEC.

### 3.3. RO-MD Process

In the RO-MD process, MD can enhance the overall water recovery rate. This section will discuss the operating parameters associated with the minimum SEC and the influence of the heat source for the MD unit on SEC. Table 2 lists the parameters of the simulation. Since the RO water recovery rate (Y_r_) is lower than 50%, Y_r_ is set to 40% or 50% in the simulation. Considering the saturated concentration of NaCl, the overall water recovery rate (Y_r_ + Y_m_) of seawater in the integrated process should be less than 85%. Moreover, the liquid entry pressure (LEP) of the MD membrane should also be taken into consideration; thus, the MD operating pressure is set to 3 bar [33]. For the estimation of the MD heat requirement, EE is set to 60% [27].

Figure 9 displays the SEC_RO-MD_ values at different water recovery rates. When the RO water recovery rate remains constant and there is sufficient waste heat as the MD heat source (Equation (15), X = 1), SEC_RO-MD_ can be reduced as the MD water recovery rate increases. When the MD water recovery rate increases from 1% to 45%, SEC_RO-MD_ decreases from 2.14 to 1.03 kWh/m^3^. If no waste heat supplies the MD process, SEC will largely increase as the water recovery rate increases due to the high energy consumption of the enthalpy of vaporization from the MD process. When the RO recovery rate is maintained at 40% and the MD water recovery rate is 45%, SEC_RO-MD_ can be as high as 584 kWh/m^3^. Because the amount of waste heat supplied by industrial sites is confined, the water recovered by the MD unit in large-scale seawater operations is less than 10% of all water recovered. Thus, both the gray and blue rectangular areas in Figure 9 reflect practical conditions.

Figure 9 indicates that waste heat can largely influence SEC_RO-MD_. If there is not sufficient waste heat supply, electricity, steam or other heat sources should be provided for the MD process. Figure 10 shows the relationship between the total water recovery rate and SEC_RO-MD_ at different percentages of the waste heat supply when the RO water recovery rate is 40%. If the waste heat could provide 90% or 95% of the required heating energy (X = 0.9 or 0.95), SEC_RO-MD_ could be reduced to 25.5 or 13.61 kWh/m^3^, respectively, when the total water recovery rate is 51%.

### 3.4. RO-MD-PRO Process

Figure 11 displays an SEC comparison between the RO-MD-PRO and RO-PRO processes. It is assumed that there is sufficient waste heat supply for MD operations. The straight line indicates SEC_RO-MD-PRO_ at different water recovery rates. This figure also displays SEC_RO-ERD_ and SEC_RO-PRO_ when the RO water recovery rate is 40%. When the RO water recovery rate is held constant, SEC_RO-MD-PRO_ decreases as the MD water recovery rate increases. Moreover, when the total water recovery rate is 64%, SEC_RO-MD-PRO_ changes from positive to negative, which suggests that energy generation by PRO is larger than energy consumption by pumping.

### 3.5. The Fractional Energy Savings(FES) of Integrated Processes

Figure 12 illustrates the FES for different water recovery rates. When the MD heat requirement is totally met by waste heat, the FES of RO-MD, RO-PRO, and RO-MD-PRO increases as the total water recovery rate increases since MD can enhance water recovery and PRO can extract osmotic pressure energy from brine discharge. At the same water production rate, when the water recovery rate increases, the amount of brine will decrease, but the brine concentration will increase. According to Equations (7) and (9), these two phenomena have opposite effects on PRO energy generation. Therefore, the increase in the rate of FES for RO-PRO is lower than that for RO-MD and RO-MD-PRO.

### 3.6. Comparison with RO-Integrated Processes in the Literature

Table 3 lists the SEC values of RO-integrated processes reported in previous studies and in this study. The results indicate that previous studies mainly focused on SEC at specific water recovery rates, and few have evaluated the impact of the MD water recovery rate on SEC. In this study, SEC at a total water recovery rate ranging from 10% to 85% was studied; moreover, the conditions associated with different MD water recovery rates were discussed. Thus, this work can provide a broader range of evaluations for process integration. Compared with Wan and Chung’s results [27], SEC_RO-PRO_ in this work is slightly higher than theirs (0.28 kWh/m^3^) when the RO water recovery rate is 50%. This difference is mainly attributed to the different efficiency settings of the pump and PX. Compared with the work of Kim et al. [25], SEC_RO-MD-PRO_ in this study is lower. The difference is that this study assumes that the MD heat source is completely provided by waste heat. When the ratio of MD water recovery to total water recovery increases, the SEC of the RO-MD-PRO process can be reduced. 

## 4. Conclusions

Based on the principles of energy consumption during fluid transport, fluid mechanical energy recovery, MD thermal energy demand and PRO energy generation, this study established theoretical SEC models for water recovery by integrating RO with other unit operations, such as MD and PRO. The purpose of this study was to analyze the effect of the total water recovery rate on SEC in different integrated processes, such as RO-PRO, RO-MD and RO-MD-PRO. From this study, the following conclusions can be drawn.

(1)The brine flow rate decreases as RO water recovery increases. When the water recovery rate is greater than 30%, recyclable mechanical energy in the PX begins decreasing in availability, and SEC starts increasing. The minimum SEC is 1.97 kWh/m^3^ at a recovery rate of 30% for the RO/ERD process.(2)For the RO-PRO process, SEC reaches a minimum value of 1.33 kWh/m^3^ at a recovery rate of 38%. RO-PRO can give a lower SEC than RO due to the energy generation by PRO, and a higher water recovery rate is obtained at the minimum SEC.(3)For the RO-MD process, when the RO water recovery rate remains constant and there is sufficient waste heat as the MD heat source, SEC can be reduced as the MD water recovery rate increases. If the water recovery rate of RO is fixed at 40% and the total water recovery rate is 85%, SEC is 1.03 kWh/m^3^.(4)For the RO-MD-PRO process and an RO water recovery rate that is constant at 40%, the energy consumption due to pumping and energy generation by PRO reaches a balance at a total water recovery rate of 64%, which means that SEC is zero under these conditions. When the total recovery rate exceeds 64%, the FES of the integrated process is greater than 100%.(5)The limit of the water recovery rate for SWRO is generally 50%. Thus, if we assume that the RO water recovery rate is 50%, the recovery rate of water for MD is 10%, and sufficient waste heat is available as a heat source for the MD unit; the SECs of the RO, RO-PRO, RO-MD and RO-MD-PRO processes are found to be 2.28, 1.47, 1.75, and 0.67 kWh/m^3^, respectively. The corresponding FES values of the integrated processes are 37%, 23% and 70% when compared with the baseline RO process.

It is noted that this study only evaluated the energy consumption for water production from integrated RO processes. When evaluating the water production cost in future engineering applications, further consideration of capital costs, such as land, construction, equipment, module, maintenance and operation costs, is needed.

## Figures and Tables

**Figure 1 membranes-12-00432-f001:**
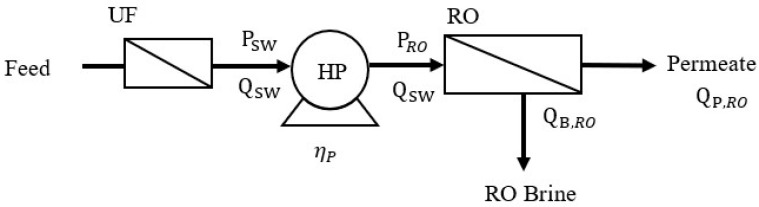
Schematic of the stand-alone RO process.

**Figure 2 membranes-12-00432-f002:**
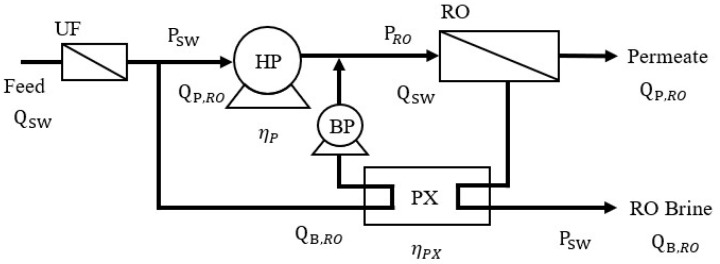
Schematic of RO with an energy recovery device (RO-ERD).

**Figure 3 membranes-12-00432-f003:**
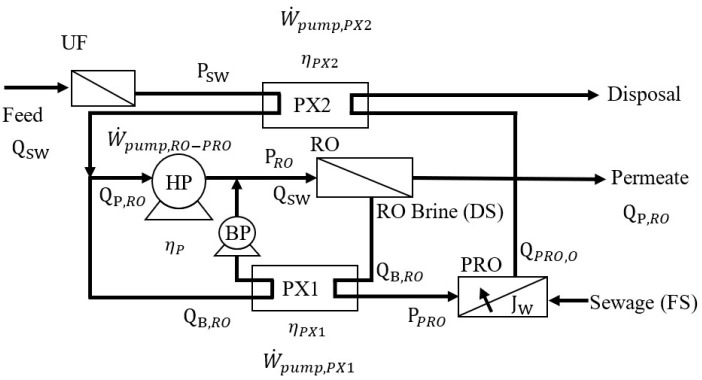
Schematic of the hybrid RO-PRO process.

**Figure 4 membranes-12-00432-f004:**
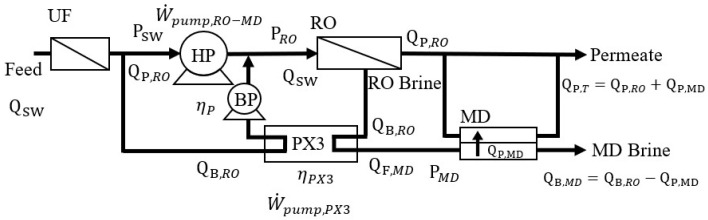
Schematic of the hybrid RO-MD process.

**Figure 5 membranes-12-00432-f005:**
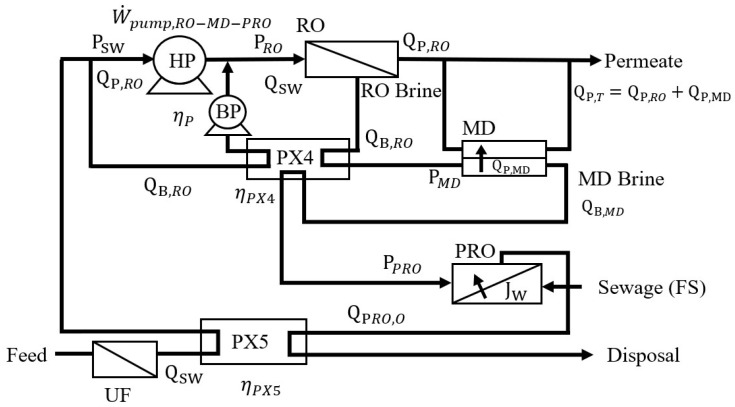
Schematic of the hybrid RO-MD-PRO process.

**Figure 6 membranes-12-00432-f006:**
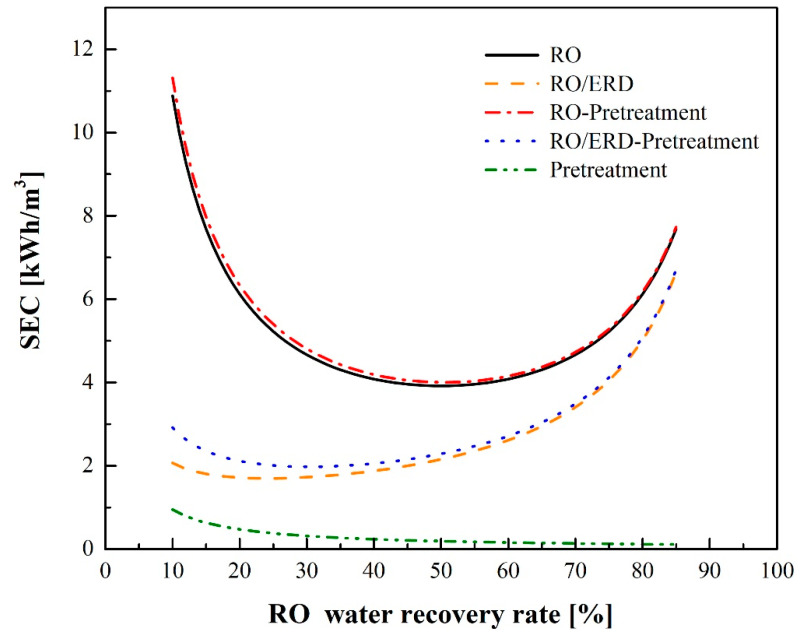
Comparison of SEC between the stand-alone RO and RO-ERD processes simulated with and without pretreatment.

**Figure 7 membranes-12-00432-f007:**
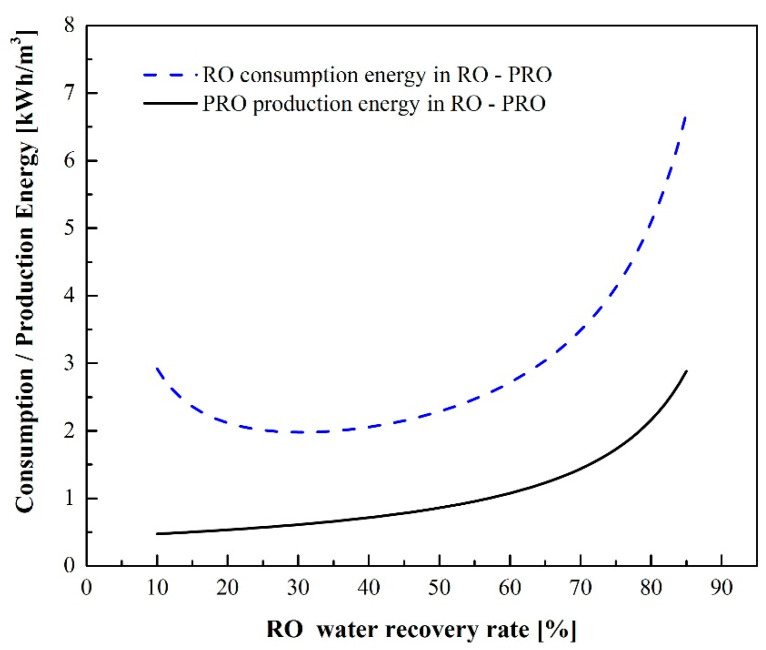
Comparison of RO energy consumption and PRO energy production in the RO-PRO process.

**Figure 8 membranes-12-00432-f008:**
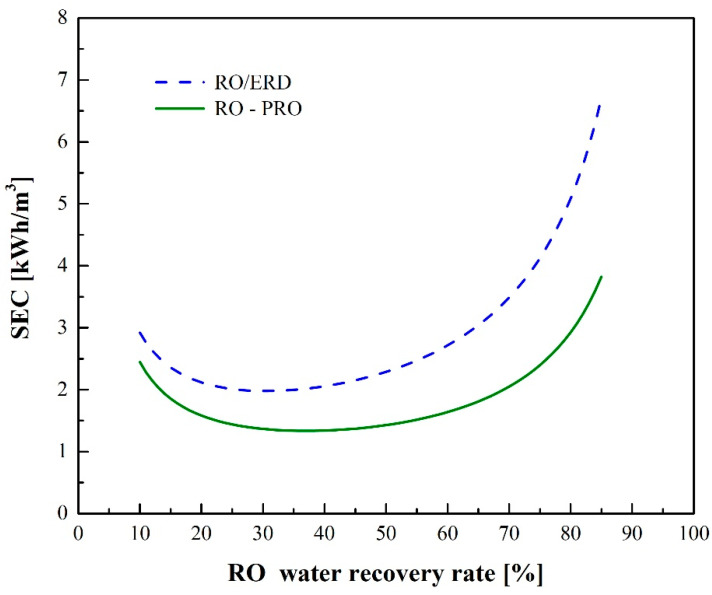
Simulation results of SE for CRO-ERD and RO-PRO at different water recovery rates.

**Figure 9 membranes-12-00432-f009:**
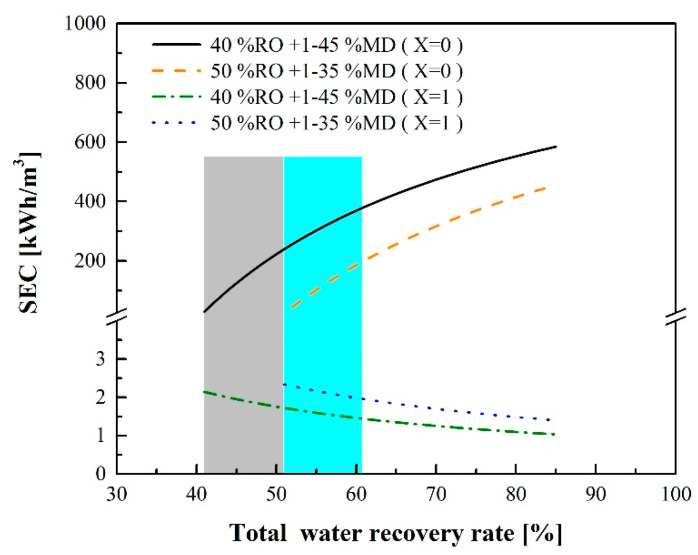
SEC of the RO-MD process with or without waste heat contribution.

**Figure 10 membranes-12-00432-f010:**
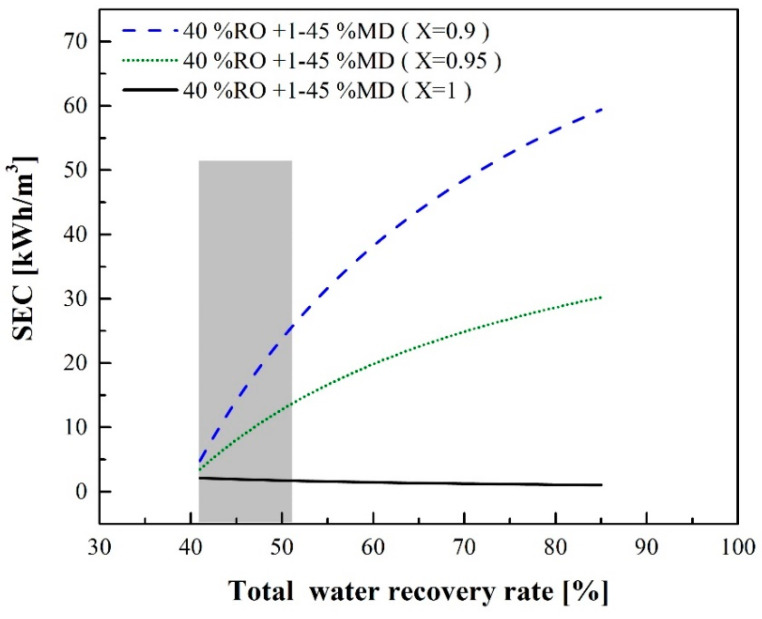
SEC of the RO-MD process with or without a sufficient waste heat source for MD.

**Figure 11 membranes-12-00432-f011:**
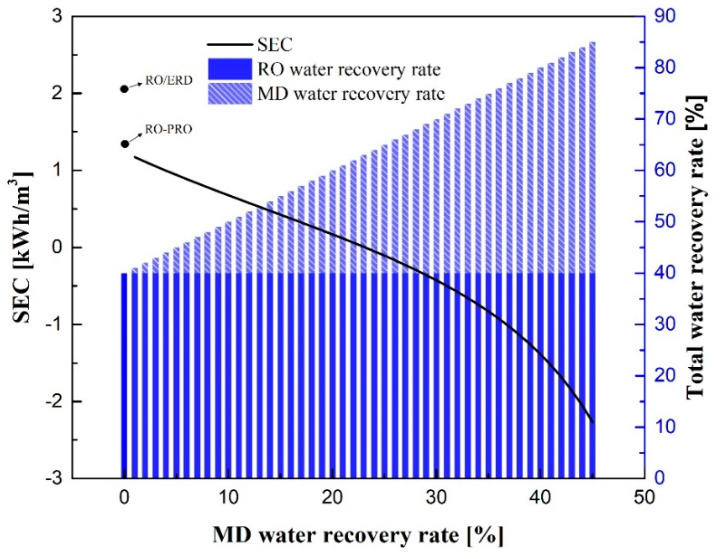
SEC of the RO-MD-PRO process at various MD water recovery rates and total water recovery rates.

**Figure 12 membranes-12-00432-f012:**
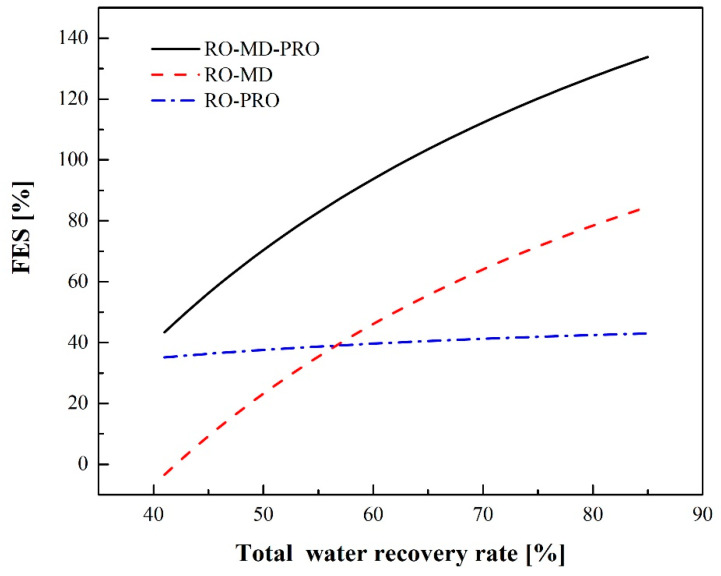
FES for different integrated processes.

**Table 1 membranes-12-00432-t001:** The parameters used in stand-alone RO and/or RO-ERD process simulations.

$C_{\mathrm{SW}}$	Seawater salt concentration (M)	0.589
$R_{t}$	Salt rejection percentage of the membrane (-)	99%
$\eta_{p}$	RO pump efficiency (-)	80%
$\eta_{\mathrm{PX}}$	Pressure exchanger efficiency (-)	95%
$Q_{P,\mathrm{RO}}$	$\mathrm{The}\;\mathrm{volumetric}\;\mathrm{flowrate}\;\mathrm{of}\;\mathrm{RO}\;\mathrm{permeate}{(m}^{3}/\mathrm{day})$	100,000

**Table 2 membranes-12-00432-t002:** Parameters for RO-MD process simulations.

$Y_{r}$	RO water recovery rate (-)	40%, 50%
$Y_{m}$	MD water recovery rate (-)	1~45%, 1~35%
$Y_{t}$	Total water recovery rate (-)	≤85%
$P_{\mathrm{MD}}$	MD pressure [4]	3
$\Delta H_{\mathrm{vap}}$	Enthalpy of vaporization (kJ⁄kg)	2382
EE	Energy efficiency of MD (-)	60%
$T_{\mathrm{FMD}}$	Temperature of the MD feed solution (°C)	70
$T_{\mathrm{BRO}}$	Temperature of the RO brine (°C)	30

**Table 3 membranes-12-00432-t003:** SEC values from the literature and simulation results in this study.

RO Process	RO-PRO Process	RO-MD Process	RO-MD-PRO Process	Ref
Y_r_ = 20, 30% SEC = 3.73, 3.38 kWh/m^3^	Y_r_ = 20, 30% SEC = 3.08, 2.64 kWh/m^3^			[34]
Y_r_ = 50% SEC = 2 kWh/m^3^	Y_r_ = 50% SEC = 1.2 kWh/m^3^			[26]
Y_r_ = 25, 50% SEC = 1.79, 2.27 kWh/m^3^	Y_r_ = 25, 50% SEC = 1.08, 1.14 kWh/m^3^			[27]
Y_r_ = 50% SEC = 1.91 kWh/m^3^	Y_r_ = 50% SEC = 1.78 kWh/m^3^		Y_r_ = 50%, Y_m_ =2% SEC = 1.60 kWh/m^3^	[25]
SEC = 3.32 kWh/m^3^	SEC = 2.869 kWh/m^3^	SEC = 2.809 kWh/m^3^	SEC = 2.683 kWh/m^3^	[35]
Y_r_ = 10 ~ 85% SEC = 1.91 ~ 8 kWh/m^3^	Y_r_ = 10 ~ 85% SEC = 1.33 ~ 4 kWh/m^3^	Y_r_ = 10 ~ 85% SEC = 1.03 ~ 2.28 kWh/m^3^	Y_r_ = 10 ~ 85% SEC = 0 ~ 1.33 kWh/m^3^	This work

## Data Availability

Data associated with this research are mentioned in the article.

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
