# Peer review of "Evaluation of the Specific Energy Consumption of Sea Water Reverse Osmosis Integrated with Membrane Distillation and Pressure–Retarded Osmosis Processes with Theoretical Models"

_membranes, 2022, doi:10.3390/membranes12040432_

Round 1

Reviewer 1 Report

This paper reports theoretical models based on energy consumption in fluid transport, fluid mechanical energy recovery, the MD thermal energy demand and PRO energy production to analyze the specific energy consumption of RO, RO-PRO, RO-MD and RO-MD-PRO systems.  Few comments below can be considered for further refining of the manuscript.

The manuscript contains a large number of abbreviations. It is necessary to make a list of abbreviations, including also subscripts and superscripts.

Lines 76-77 - Correct index kWh/m3

In the last two paragraphs of the Introduction, there are statements that are listed using the numerical designation (1), (2), etc. It is necessary to remove the numerical designation or replace it with a letter designation. Otherwise, these notations can be confused with the numbers of equations.

Table 1 has different font sizes.

Reviewer 2 Report

The manuscript titled “Evaluation of the Specific Energy Consumption of Sea Water Reverse Osmosis Integrated with Membrane Distillation and Pressure Retarded Osmosis Processes with theoretical models” and written by Shao-Chi Tsai et al. has some drawbacks. I recommend a major revision based on the following comments:

  1. Please, include a reference that support the statement “This integrated approach can reduce the energy requirement for water 42 recovery and enhance the water recovery rate” in page 1, line 42.
  2. Page 2, line 50 it is written “In an MD…” maybe “In a MD…
  3. Page 2, between line 67 and 78, the authors commented some works related with specific energy consumption in SWRO, SWRO with PRO and or MD. This is ok, but, there are missing relevant published studies regarding performance assessment of SWRO spiral-wound membrane modules and SWRO systems under variable operating conditions that should be included.
  4. Page 2, line 77, number 3 should be written as superscript, Please, revise the entire manuscript.
  5. Page 3. According with the Figure 1, in Equation 1, should not be permeate flow instead of water recovery? Usually, the water recovery is expressed in percentage and the Equation 1 is usually called specific energy consumption. Why this nomenclature? Please, write h instead of hr. Revise the manuscript.
  6. The Equation are hard to follow, Why the salt rejection is in Eq. 3? Actually, I do not understand Equation 3 neither the sequence of the equation. I mean, to calculate the energy consumption per m3 in the different systems (RO; RO-PRO, etc) is relatively simple, just calculate the powered needed by all pumps divided by permeate flow. The only additional parameter is just taking into consideration the energy recovery device. I suggest the authors to reformulate the equations, if the authors want to make a more precise energy calculations, I recommend using enthalpy balance in the different devices.
  7. Did the authors consider the pumps of sewage (FS) and feed in Figure 3? Pressure drop in the ultrafiltration? And the pumping from first PX! To the PRO system? PX1 is pressurizing for the feed of RO or the draw solution of PRO, it seems the feed of RO but then something is missing in the PRO system.
  8. In Figure 5, PX4 has three input and three outputs? Could the authors provide the specification of this equipment? As much as I know pressure exchanger work with 2 flows, not 3.
  9. In Table 1 there are different font sizes.
  10. Figure 6. SEC does not only depends on flux recovery, it also depends on feed pressure and feed flow, and there are multiple operating points, for example see the paper: Processes 2020, 8(6), 692; https://doi.org/10.3390/pr8060692
  11. Regarding the result in page 9, line 246. RO-PRO had a SEC of 1.33 kWh/m3 and RO a SEC of 2 kwh/m3, so, the PRO system was able to generate close to 0.7 kWh/m3. Could the authors provide any reference where a PRO system could generate such amount of energy?
  12. There is a typo in page 10, line 265, I think it is equation 15 instead of 16.
  13. The integration f MD is tricky, it provides only good results when there is waste heat to be used, this is not fair comparison. MD allow increasing flux recovery but at high SEC requirements

Round 2

Reviewer 2 Report

Authors have adressed all my comments